# Opposite microglial activation stages upon loss of PGRN or TREM2 result in reduced cerebral glucose metabolism

Julia K Götzl[1,†], Matthias Brendel[2,†], Georg Werner[1,†], Samira Parhizkar[1], Laura Sebastian Monasor[3], Gernot Kleinberger[1,4] (iD), Alessio-Vittorio Colombo[3], Maximilian Deussing[2], Matias Wagner[5,6,7], Juliane Winkelmann[5,6,7], Janine Diehl-Schmid[8], Johannes Levin[3,9], Katrin Fellerer[1], Anika Reifschneider[1], Sebastian Bultmann[10], Peter Bartenstein[2,4], Axel Rominger[2,4], Sabina Tahirovic[3] (iD), Scott T Smith[11], Charlotte Madore[11], Oleg Butovsky[11,12], Anja Capell[1,*] (iD) & Christian Haass[1,3,4,**] (iD)

## Abstract

Microglia adopt numerous fates with homeostatic microglia (HM) and a microglial neurodegenerative phenotype (MGnD) representing two opposite ends. A number of variants in genes selectively expressed in microglia are associated with an increased risk for neurodegenerative diseases such as Alzheimer's disease (AD) and frontotemporal lobar degeneration (FTLD). Among these genes are progranulin (*GRN*) and the triggering receptor expressed on myeloid cells 2 (*TREM2*). Both cause neurodegeneration by mechanisms involving loss of function. We have now isolated microglia from $Grn^{-/-}$ mice and compared their transcriptomes to those of $Trem2^{-/-}$ mice. Surprisingly, while loss of *Trem2* enhances the expression of genes associated with a homeostatic state, microglia derived from $Grn^{-/-}$ mice showed a reciprocal activation of the MGnD molecular signature and suppression of gene characteristic for HM. The opposite mRNA expression profiles are associated with divergent functional phenotypes. Although loss of TREM2 and progranulin resulted in opposite activation states and functional phenotypes of microglia, FDG (fluoro-2-deoxy-D-glucose)-μPET of brain revealed reduced glucose metabolism in both conditions, suggesting that opposite microglial phenotypes result in similar wide spread brain dysfunction.

**Keywords** disease-associated and homeostatic microglial signatures; microglia; neurodegeneration; progranulin; TREM2
**Subject Categories** Immunology; Metabolism; Neuroscience

## Introduction

While for a long time researchers distinguished only two distinct stages of microglia, the M1 and the M2 phenotype, recent evidence strongly indicates that a multitude of functionally diverse microglial populations exists in a dynamic equilibrium (Ransohoff, 2016; Keren-Shaul *et al*, 2017; Krasemann *et al*, 2017). This becomes very apparent when one compares mRNA signatures of microglia isolated from various mouse models for neurodegeneration and compares them to controls (Abduljaleel *et al*, 2014; Butovsky *et al*, 2015; Holtman *et al*, 2015; Keren-Shaul *et al*, 2017; Krasemann *et al*, 2017). In mouse models for neurodegenerative diseases, mRNA signatures were identified which are characteristic for a disease-associated microglia (DAM)/a microglial neurodegenerative phenotype (MGnD) whereas in controls a homeostatic microglial (HM) signature was observed (Butovsky *et al*, 2014, 2015; Holtman *et al*,

1   Chair of Metabolic Biochemistry, Biomedical Center (BMC), Faculty of Medicine, Ludwig-Maximilians-Universität München, Munich, Germany
2   Department of Nuclear Medicine, University Hospital, Ludwig-Maximilians-Universität München, Munich, Germany
3   German Center for Neurodegenerative Diseases (DZNE), Munich, Germany
4   Munich Cluster for Systems Neurology (SyNergy), Munich, Germany
5   Institut für Neurogenomik, Helmholtz Zentrum München, Munich, Germany
6   Institut of Human Genetics, Technische Universität München, Munich, Germany
7   Institute of Human Genetics, Helmholtz Zentrum München, Neuherberg, Germany
8   Department of Psychiatry, Technische Universität München, Munich, Germany
9   Department of Neurology, University Hospital, Ludwig-Maximilians-Universität München, Munich, Germany
10  Department of Biology and Center for Integrated Protein Science Munich (CIPSM), Ludwig Maximilians-Universität München, Munich, Germany
11  Ann Romney Center for Neurologic Diseases, Department of Neurology, Brigham and Women's Hospital, Harvard Medical School, Boston, MA, USA
12  Evergrande Center for Immunologic Diseases, Brigham and Women's Hospital, Harvard Medical School, Boston, MA, USA
    *Corresponding author. Tel: +4989440046535; E-mail: anja.capell@mail03.med.uni-muenchen.de
    **Corresponding author. Tel: +4989440046550; E-mail: christian.haass@mail03.med.uni-muenchen.de
    †These authors contributed equally to this work

2015; Keren-Shaul *et al*, 2017; Krasemann *et al*, 2017). MGnD upregulates a characteristic set of genes, which may initially allow microglia to respond to neuronal injury in a defensive manner. This includes the induction of pathways triggering phagocytosis, chemotaxis/migration, and cytokine release. The upregulation of genes involved in those pathways goes along with a suppression of homeostatic genes (Butovsky *et al*, 2015; Krasemann *et al*, 2017). Key genes involved in the switch of HM to MGnD are regulated by the TREM2 (triggering receptor expressed on myeloid cells 2) ApoE (apolipoprotein E) pathway (Krasemann *et al*, 2017).

A pivotal role of microglia in neurodegeneration is strongly supported by the identification of sequence variants found in a number of genes robustly or even selectively expressed within microglia in the brain, among them *GRN* (encoding the progranulin (PGRN) protein; Baker *et al*, 2006; Cruts & Van Broeckhoven, 2008; Zhang *et al*, 2014; Gotzl *et al*, 2016; Lui *et al*, 2016) and *TREM2* (Guerreiro *et al*, 2013; Jonsson & Stefansson, 2013; Jonsson *et al*, 2013; Rayaprolu *et al*, 2013; Borroni *et al*, 2014; Cuyvers *et al*, 2014; Ulrich *et al*, 2017). Mutations in the *GRN* gene (Baker *et al*, 2006; Cruts *et al*, 2006) cause frontotemporal lobar degeneration (FTLD) with TDP-43 (TAR-DNA binding protein 43)-positive inclusions due to haploinsufficiency. PGRN is a growth factor-like protein with neurotrophic properties in the brain (Van Damme *et al*, 2008). PGRN is also transported to lysosomes (Hu *et al*, 2010; Zhou *et al*, 2015) where it appears to regulate expression and activity of lysosomal proteins (Ahmed *et al*, 2010; Hu *et al*, 2010; Wils *et al*, 2012; Tanaka *et al*, 2013a,b; Gotzl *et al*, 2014, 2016, 2018; Beel *et al*, 2017; Chang *et al*, 2017; Ward *et al*, 2017; Zhou *et al*, 2017). PGRN is proteolytically processed into granulin peptides, which can be found in biological fluids (Bateman *et al*, 1990; Shoyab *et al*, 1990; Belcourt *et al*, 1993; Cenik *et al*, 2012). TREM2 is produced as a membrane-bound type-1 protein (Kleinberger *et al*, 2014), which traffics to the cell surface where it mediates signaling via binding to its co-receptor, the DNAX activation protein of 12 kDa (DAP12; Ulrich & Holtzman, 2016; Yeh *et al*, 2017). Signaling is terminated by proteolytic shedding of the TREM2 ectodomain (Kleinberger *et al*, 2014; Schlepckow *et al*, 2017). Several sequence variants associated with TREM2 cause neurodegeneration via a loss of function (Kleinberger *et al*, 2014, 2017; Schlepckow *et al*, 2017; Ulland *et al*, 2017; Song *et al*, 2018). Sequence variants of TREM2 affect a multitude of functions including chemotaxis, migration, survival, binding of phospholipids and ApoE, proliferation, survival, and others (Kleinberger *et al*, 2014, 2017; Atagi *et al*, 2015; Bailey *et al*, 2015; Wang *et al*, 2015; Yeh *et al*, 2016; Mazaheri *et al*, 2017; Ulland *et al*, 2017). Strikingly, a loss of TREM2 function locks microglia in a homeostatic state (Krasemann *et al*, 2017; Mazaheri *et al*, 2017). Instead of suppressing their homeostatic mRNA signature like mouse models of neurodegenerative disorders, in the absence of TREM2 microglia even enhance expression of homeostatic genes and fail to express the disease-associated signature (Krasemann *et al*, 2017; Mazaheri *et al*, 2017). As a result, TREM2 deficiency decreases chemotaxis, phagocytosis, and barrier function (Kleinberger *et al*, 2017; Mazaheri *et al*, 2017; Ulland *et al*, 2017). TREM2 therefore appears to play a key role as a central hub gene in the regulation of microglial homeostasis. We now investigated microglial gene expression and function in the absence of PGRN and made the surprising observation that loss of TREM2 or PGRN leads to opposite microglial activity phenotypes, which, however, both cause wide spread brain dysfunction.

# Results

## Opposite molecular signatures of microglia in *Grn*$^{-/-}$ and *Trem2*$^{-/-}$ mice

Loss-of-function mutations in *TREM2* are associated with various types of neurodegeneration, including a FTLD-like syndrome (Ulrich & Holtzman, 2016). Similarly, haploinsufficiency of *GRN* is associated with TDP-43-positive FTLD (Baker *et al*, 2006; Cruts *et al*, 2006; Cruts & Van Broeckhoven, 2008). At least some of the *GRN*-dependent FTLD-associated phenotypes can be mimicked in a mouse model entirely lacking PGRN (Ahmed *et al*, 2010; Yin *et al*, 2010; Wils *et al*, 2012; Gotzl *et al*, 2014). Furthermore, both, a *Trem2* knockout and the knockin of the p.T66M mutation, mimic features of a FTD-like syndrome (Kleinberger *et al*, 2017; Mazaheri *et al*, 2017). Since both proteins are preferentially expressed in microglia, we compared loss-of-PGRN-associated microglial phenotypes with those known for TREM2 deficiency (Krasemann *et al*, 2017; Mazaheri *et al*, 2017). To do so, we first purified microglia from brains of adult *Grn*$^{-/-}$ mice by fluorescence-associated cell sorting (FACS) using microglia-specific anti-FCRLS and anti-CD11b antibodies. We then investigated the expression pattern of gene characteristic for MGnD and HM using NanoString gene expression profiling (MG534; Butovsky *et al*, 2014; Krasemann *et al*, 2017; Mazaheri *et al*, 2017; Dataset EV1). Gene expression levels in each sample were normalized against the geometric mean of five housekeeping genes including *Cltc*, *Gapdh*, *Gusb*, *Hprt1*, and *Tubb5*. Out of 529 genes analyzed, 58 genes were significantly upregulated and 58 genes were downregulated. Strikingly, genes most strongly upregulated in *Grn*$^{-/-}$ microglia are those previously described for MGnD (Fig 1A; Butovsky *et al*, 2015; Holtman *et al*, 2015; Keren-Shaul *et al*, 2017; Krasemann *et al*, 2017). These include *ApoE*, as the most upregulated gene, *Ly9*, *Clec7a*, *Dnajb4*, *Ccl4*, and many others suggesting that PGRN-deficient microglia adopt the MGnD state. We then compared the *Grn*$^{-/-}$ microglial signature to the previously analyzed molecular signature of *Trem2*$^{-/-}$ microglia (Fig 1B and C; Mazaheri *et al*, 2017). Both NanoString gene expression panels overlapped in 418 genes. Out these, 359 mRNAs could be detected in both screens (Dataset EV1). In the *Grn*$^{-/-}$ microglia, 40 mRNAs were upregulated and 55 mRNAs were downregulated, whereas in the *Trem2*$^{-/-}$ microglia, 87 mRNAs were increased and 27 mRNAs were decreased (Fig 1C; left two panels). Interestingly, while only six mRNAs were equally up/downregulated in both phenotypes, 32 mRNAs showed an opposite regulation (Fig 1C; right panel). Strikingly, while in *Grn*$^{-/-}$ mice the neurodegenerative disease-associated signature is massively upregulated, this set of genes is suppressed in the *Trem2*$^{-/-}$ microglia (Fig 1B and D; Mazaheri *et al*, 2017). Similarly, the homeostatic mRNA signature is slightly but significantly upregulated in *Trem2*$^{-/-}$ microglia (Mazaheri *et al*, 2017) but severely suppressed in *Grn*$^{-/-}$ microglia (Fig 1B and D).

## Confirmation of molecular microglial signatures on protein level

Expression of proteins associated with MGnDs such as ApoE, CLEC7A, TREM2, and CD68 was also increased in acutely isolated microglia (Fig 2A and B). sTREM2 was also found to be increased in brains and serum of *Grn*$^{-/-}$ mice (Fig 2C and D), although no increase in *Trem2* mRNA levels was observed in the NanoString analysis (Fig 1B), suggesting posttranscriptional regulation.

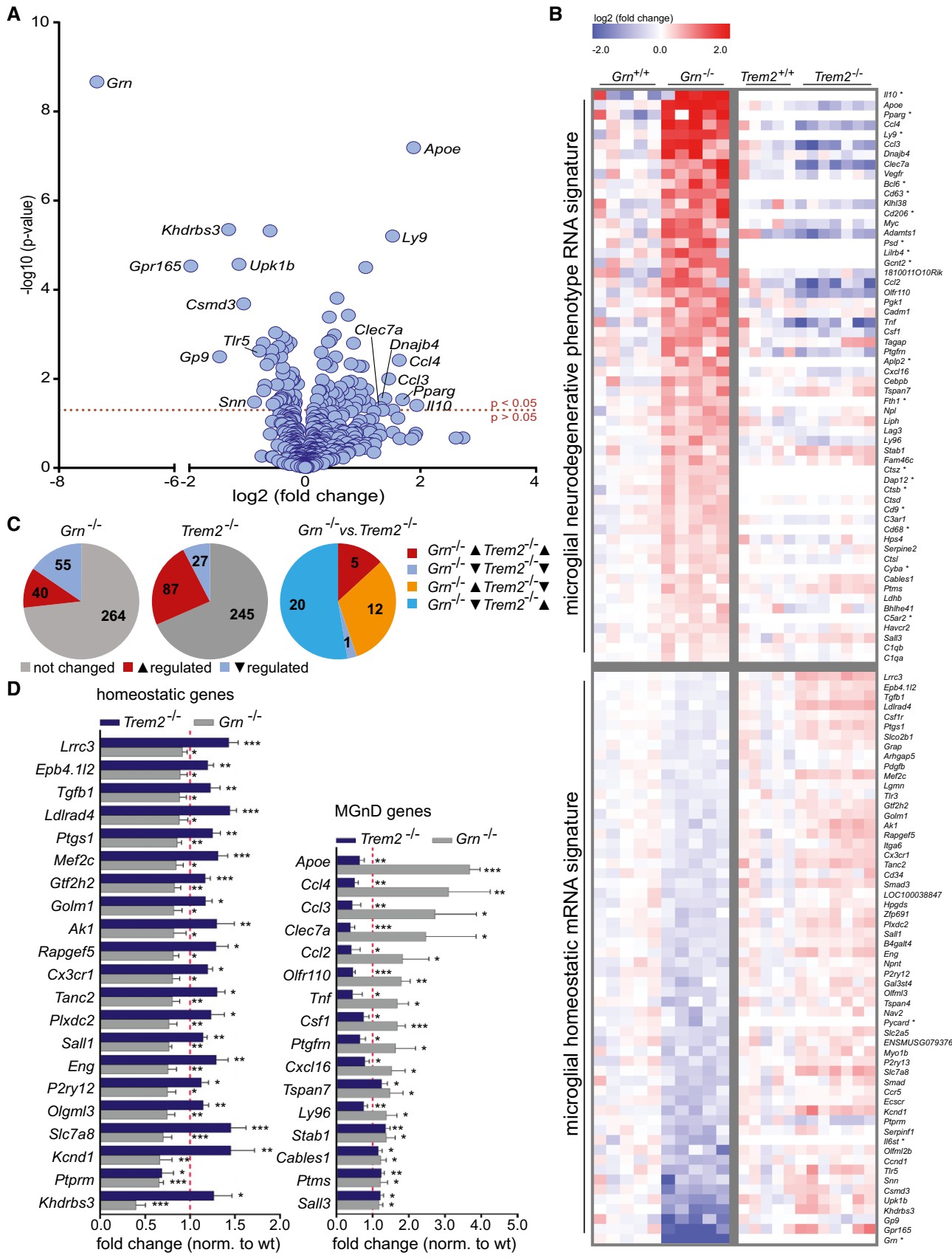

**Figure 1.**

**Figure 1. Opposite mRNA signatures of $Grn^{-/-}/Trem2^{+/+}$ and $Trem2^{-/-}/Grn^{+/+}$ microglia.**

A  Volcano blot representation of the differently expressed genes in FCRLS- and CD11b-positive $Grn^{-/-}$ microglia in comparison with wt microglia isolated from 5.5-month-old mice (male, n = 5). 116 genes out of 529 genes analyzed are significantly changed, and from these, 58 genes are upregulated and 58 genes are downregulated. Eight up- and down regulated genes with the highest fold change are indicated.

B  Heatmap of significantly affected genes (P < 0.05) in FCRLS- and CD11b-positive $Grn^{-/-}$ microglia in comparison with $Grn^{+/+}$ microglia isolated from 5.5-month-old mice (n = 5 per genotype). For the significantly affected genes of the $Grn^{-/-}$ microglia, mRNA expression data of the $Trem2^{-/-}$ microglia in comparison with the corresponding wt microglia were taken from previously published data (Mazaheri et al, 2017). The RNA counts for each gene and sample were normalized to the mean value of wt followed by a log2 transformation (n ≥ 5 per genotype). *labeled genes were not analyzed or below detection limit in $Trem2^{-/-}$ microglial mRNA expression dataset (Mazaheri et al, 2017).

C  Changes in gene expression of 359 detected genes in $Grn^{-/-}$ microglia and $Trem2^{-/-}$ microglia. Note that from the 38 genes significantly altered in both genotypes, 32 genes are regulated in opposite direction.

D  Expression levels of significantly altered homeostatic and MGnD genes of $Grn^{-/-}$ and $Trem2^{-/-}$ microglia from the data set in (B). Gene expression is normalized to the mean of the wt cohort in comparisons with the published normalized dataset of $Trem2^{-/-}$ microglia (Mazaheri et al, 2017). Data represent the mean +/− SD.

Data information: For statistical analysis, unpaired two-tailed Student's t-test was performed between $Grn^{-/-}$ microglia in comparison with $Grn^{+/+}$ microglia or $Trem2^{-/-}$ microglia in comparison with $Trem2^{+/+}$ microglia: not significant P > 0.05; *P < 0.05; **P < 0.01; and ***P < 0.001.

Furthermore, protein expression of the homeostatic *P2ry12* gene is downregulated in cortical $Grn^{-/-}$ microglia (Fig 2E and F), while cortical microglia from $Trem2^{-/-}$ mice show strongly elevated P2RY12 levels (Fig 2E and F), consistent with the finding that microglia from both mouse models are in opposite activation states. These findings were further confirmed in genetically modified BV2 microglia-like cells lacking PGRN expression, where ApoE, CLEC7A, and TREM2 were upregulated, whereas P2RY12 was downregulated (Fig 2G–I). In line with the protein expression, in PGRN-deficient BV2 cells, transcripts encoding *Apoe*, *Clec7a*, and *Trem2* were also significantly upregulated, whereas *P2ry12* was downregulated (Fig 2J) further confirming that loss of PGRN results in a microglial hyperactivation.

### Increased phagocytic capacity, chemotaxis, and clustering around amyloid plaques upon loss of PGRN

Next, we investigated a series of functional phenotypes in *Grn* loss-of-function mutants, which may be differentially affected by dysregulated *Trem2* or *Grn* gene expression. First, we investigated the phagocytic capacity of PGRN-deficient BV2 cells, which recapitulate expression profile changes found in isolated $Grn^{-/-}$ microglia such as enhanced expression of ApoE, CLEC7A, and TREM2 (Fig 2). We observed a significantly increased uptake of pHrodo-labeled *Escherichia coli* (*E. coli*) in PGRN-deficient BV2 cells compared to wt

(Fig 3A). Enhanced uptake of pHrodo-labeled bacteria is in strong contrast to the reduced uptake detected in *Trem2* loss-of-function mutations (Fig EV1A; Kleinberger et al, 2014, 2017; Schlepckow et al, 2017), thus demonstrating that differentially regulated mRNA signatures translate into opposite phagocytic phenotypes. In acutely isolated microglia, uptake of bacteria was also reduced upon loss of TREM2 (Fig EV1B). $Grn^{-/-}$ microglia displayed only a slight but significant increase in phagocytosis (Fig EV1B). The rather minor effect of the *Grn* knockout on increased microglial phagocytosis is most likely due to isolation-induced activation of microglia (Gosselin et al, 2017), which as a consequence already show a rather high uptake capacity.

Previously, we demonstrated that microglia of $Trem2^{-/-}$ mice are locked in a homeostatic state and fail to migrate (Mazaheri et al, 2017). Since microglia of $Grn^{-/-}$ mice express a RNA profile typical for MGnD, we investigated if that results in increased migration. To assess migration *ex vivo*, we cultured organotypic brain slices from young (P6-7) mice as recently described (Daria et al, 2017; Mazaheri et al, 2017). In line with our previous findings, we found only baseline migration of wt microglia reflected by few migrating CD68-positive cells detected mainly in the vicinity of the brain tissue (Fig 3B–D). On the contrary, the number of migrating CD68-positive cells and their distance migrated were both significantly increased in brain slices derived from the $Grn^{-/-}$ mice (Fig 3B–D). Again, this finding is in clear contrast to our previous observation in cultured

**Figure 2. Expression of microglial marker protein characteristic for the homeostatic or disease-associated state.**

A  Immunoblot analysis of ApoE, CLEC7A, TREM2, and CD68 in lysates of acutely isolated microglia from $Grn^{-/-}$ and $Grn^{+/+}$ mice (9 months of age, n = 3 per genotype, female). IBA1 was used as loading control. The asterisk indicates an unspecific band.

B  Quantification of protein expression normalized to levels of $Grn^{+/+}$ microglia (n = 3). Data represent the mean ± SD.

C  ELISA-mediated quantification of sTREM2 in brain homogenates of $Grn^{-/-}$ and $Grn^{+/+}$ mice. Data are shown as mean ± SEM (n = 3–6).

D  ELISA-mediated quantification of sTREM2 in serum of $Grn^{-/-}$ and $Grn^{+/+}$ mice. Data are shown as mean ± SEM (n = 4–13).

E  Microglial expression of P2RY12 in cortical sections of wt, $Grn^{-/-}$ and $Trem2^{-/-}$ mice. (9 months of age, female). Scale bar indicates 10 μm.

F  Quantification of P2RY12-positive microglia. Data are shown as mean ± SD (n = 3 per genotype, female, except one male for $Trem2^{-/-}$).

G  Immunoblot analysis of secreted PGRN and ApoE (ApoE$_{med}$) in conditioned media and PGRN, ApoE, CLEC7A, P2RY12, and TREM2 in lysates of BV2 wild-type ($Grn_{wt}$) and PGRN-deficient ($Grn_{mut}$) cells. Soluble amyloid precursor protein (APPs) and calnexin were used as a loading control. The asterisk indicates an unspecific band.

H  Quantification of immunoblots normalized to BV2 $Grn_{wt}$ levels (n = 3–5). Data represent the mean ± SD.

I  ELISA-mediated quantification of sTREM2 in conditioned media of BV2 $Grn_{wt}$ and $Grn_{mut}$ cells. Data represent the mean ± SD (n = 4).

J  Quantification of relative mRNA levels of *Apoe*, *Clec7a*, *P2ry12*, and *Trem2* in BV2 $Grn_{wt}$ and $Grn_{mut}$ cells. Data represent the mean ± SD (n = 3).

Data information: For statistical analysis, unpaired two-tailed Student's t-test was performed between $Grn_{mut}$ or $Grn^{-/-}$ in comparison with $Grn_{wt}$ or $Grn^{+/+}$: n.s., not significant; *P < 0.05; **P < 0.01; ***P < 0.001; and ****P < 0.0001. For (F), the one-way ANOVA with Tukey post hoc test was used.

Source data are available online for this figure.

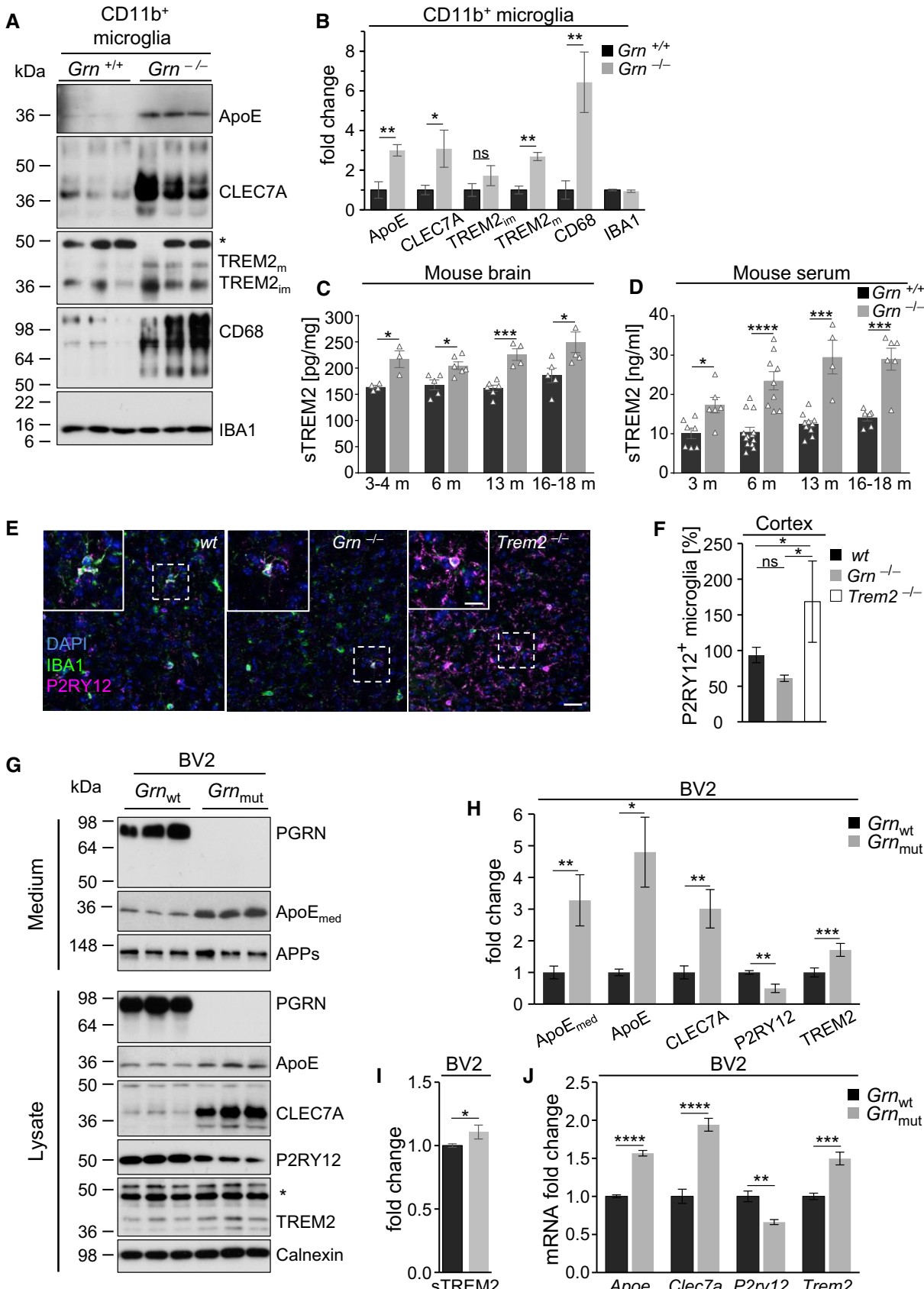

**Figure 2.**

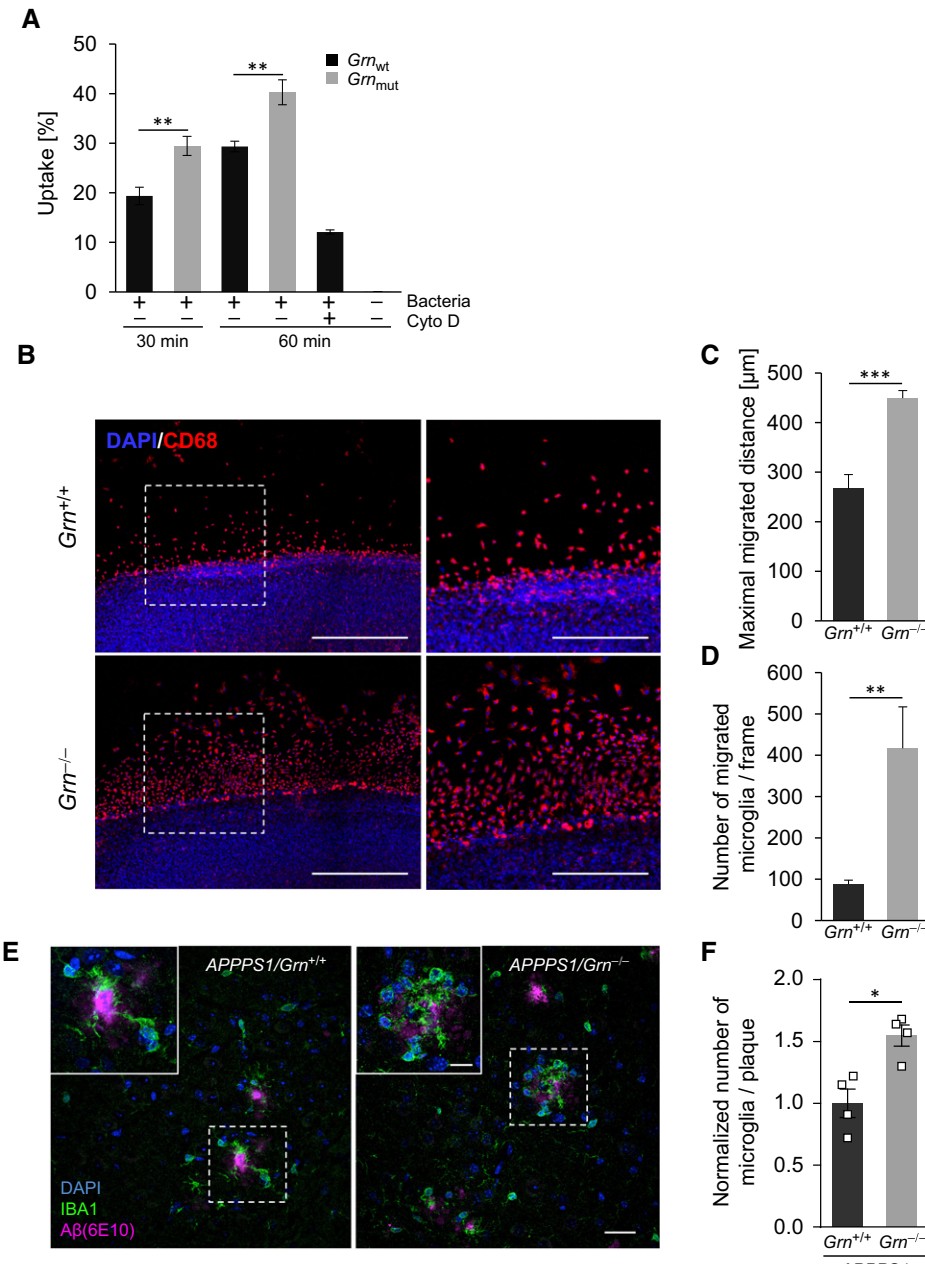

**Figure 3. Enhanced phagocytosis, migration, and clustering around amyloid plaques upon PGRN deficiency.**

A   Flow cytometric analysis of phagocytic capacity in BV2$_{wt}$ and BV2$_{mut}$ cells using pHrodo green *E. coli* as target particles. Phagocytosis was terminated after 30 and 60 min (min) of incubation. Data are presented as mean percentage of cells positive for pHrodo uptake $\pm$ SD ($n = 3$).

B   Enhanced migration of CD68-positive, PGRN-deficient cells in an *ex vivo* model. Immunofluorescence analysis of cultured *Grn$^{+/+}$* and *Grn$^{-/-}$* organotypic brain at 7 DIV immunostained using microglial marker CD68. Nuclei were counterstained using DAPI. Images of boxed regions in left panels are shown at higher magnifications in right panels. Scale bars: 500 μm (left panels), 250 μm (right panels).

C   Quantitative analysis reveals that *Grn$^{-/-}$* microglia migrate larger distances than *Grn$^{+/+}$*. Data are shown as mean $\pm$ SD ($n = 3$ independent experiments).

D   Quantification displays an increased number of migrating *Grn$^{-/-}$* cells compared to *Grn$^{+/+}$* cells. Data are shown as mean $\pm$ SD ($n = 3$ independent experiments).

E   Enhanced microglial clustering around amyloid plaques in the absence of PGRN. Left: IBA1-stained microglial clustering around X-34-positive amyloid plaque cores in 4-month-old APPPS1/*Grn$^{+/+}$* mice. Right: Age-matched APPPS1/*Grn$^{-/-}$* mice display microglial hyper-clustering around amyloid plaques compared with APPPS1/*Grn$^{+/+}$*. Dotted white boxes indicate the area that is shown at higher magnification. Scale bars indicate 10 and 3 μm in inset.

F   Immunohistochemical quantification of IBA1-positive microglial cell number per plaque (APPPS1/*Grn$^{+/+}$* $n = 4$, APPPS1/*Grn$^{-/-}$* $n = 4$; 4-month-old; two males and two females per genotype) normalized to APPPS1/*Grn$^{+/+}$* mice. Approximately 20–50 amyloid plaques were counted per mouse in 3 cortical sections to calculate the number of IBA1-positive microglia per plaque. Data are shown as mean $\pm$ SD.

Data information: For statistical analysis, unpaired two-tailed Student's *t*-test was performed (A, C, D) between PGRN-deficient Grn$_{mut}$ BV2 cells or *Grn$^{-/-}$* microglia and Grn$_{wt}$ BV2 cells or *Grn$^{+/+}$* microglia. For (F) the Mann–Whitney *U*-test, two-tailed analysis was used. Significance is indicated by *$P < 0.05$; **$P < 0.01$; and ***$P < 0.001$. Source data are available online for this figure.

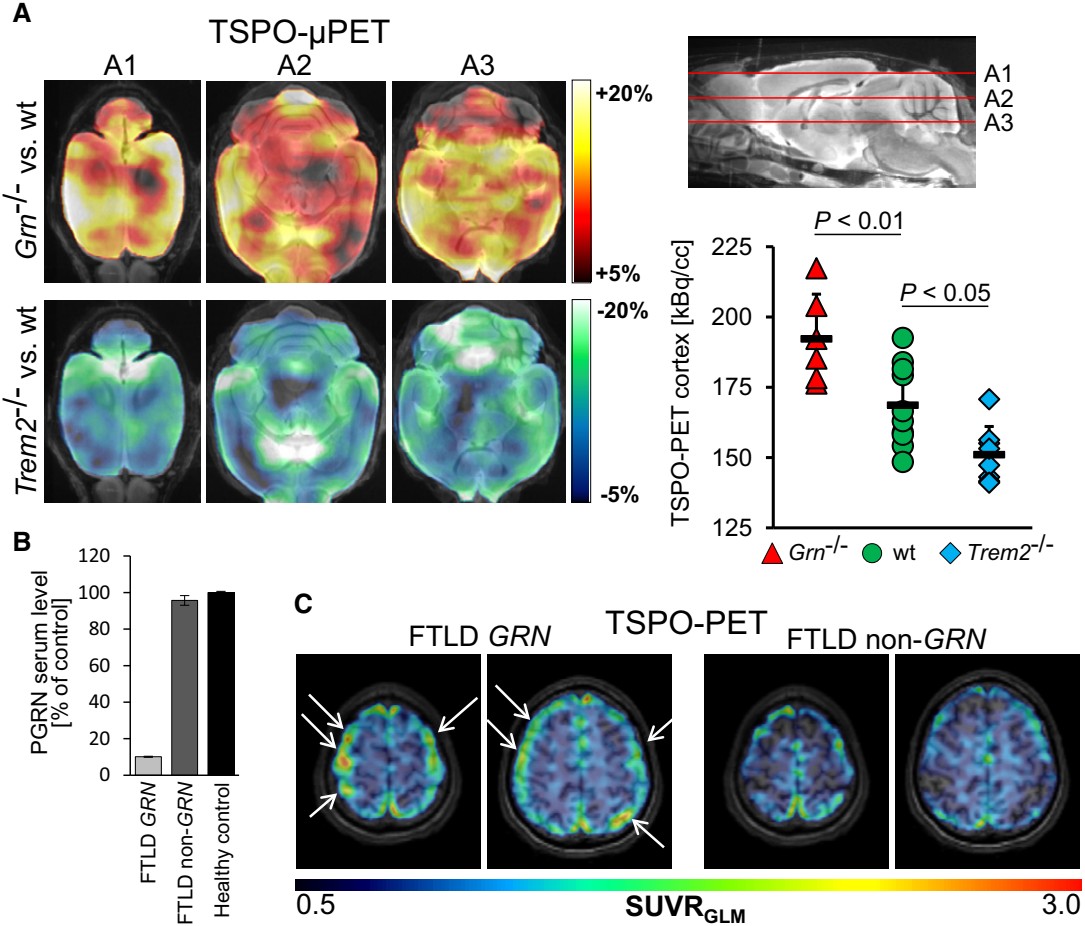

**Figure 4. Enhanced microglial activity in Grn⁻/⁻ mice and in a patient with Grn haploinsufficiency.**

A   TSPO-μPET in rodents: Axial slices (A1, A2, A3) of averaged %-TSPO-PET differences between $Grn^{-/-}$ or $Trem2^{-/-}$ mice and wt indicate increased microglial activity in the cerebrum of $Grn^{-/-}$ mice (hot color scale) and decreased microglial activity in the cerebrum of $Trem2^{-/-}$ mice (cold color scale). Scatter plot depicts single TSPO-μPET values deriving from a neocortical volume of interest. Data represent the mean +/− SD. μPET results are illustrated upon an MRI template; 6–10 female mice per group ($Grn^{-/-}$ $n$ = 6 at 8 months, $Trem2^{-/-}$ $n$ = 8 at 11 months; wt $n$ = 10 at 8–11 months). Statistics were derived from one-way ANOVA with Tukey post hoc test.

B   PGRN serum levels of the patients subjected to TSPO-PET confirmed the $GRN$ mutation and non-$GRN$ mutation carrier. PGRN was measured by ELISA in technical triplicates and normalized to serum levels of a healthy control. Data represent the mean +/− SD.

C   TSPO-PET in human FTLD patients: Visual interpretation of axial TSPO-PET SUVR images in upper cranial layers reveals increased microglial activity in frontal and parietal cortices (white arrows) of a FTLD patient with a $GRN$ loss-of-function mutation in comparison with a FTLD patient without a $GRN$ mutation. PET results are illustrated upon an MRI template; 60–80 min p.i. emission recording; and global mean scaling. Both patients were low-affinity binder.

organotypic brain slices derived from $Trem2^{-/-}$ mice, where we found a severe inhibition of migration as compared to wt (Mazaheri *et al*, 2017). The increased migratory potential is in line with enhanced clustering of IBA1-positive microglia that we detected around amyloid plaques in APPPS1/$Grn^{-/-}$ versus APPPS1/$Grn^{+/+}$ mice (Figs 3E and F, and EV1C). Again, this finding is opposite to what was found in the absence of functional TREM2, namely impaired clustering of microglia around amyloid plaques (Wang *et al*, 2016; Parhizkar *et al*, 2019) and a general reduction of chemotaxis (Mazaheri *et al*, 2017).

### Increased TSPO-PET signals in Grn⁻/⁻ mice and a human patient with GRN haploinsufficiency

To confirm that our findings have relevance *in vivo*, we searched for differences in microglial activity between $Trem2^{-/-}$ and $Grn^{-/-}$ mice by previously established *in vivo* TSPO-μPET imaging (Liu

*et al*, 2015; Kleinberger *et al*, 2017). In line with the hypothesis gained from *in vitro* experiments, we observed a striking upregulation of microglial activity throughout the cerebrum of $Grn^{-/-}$ mice when compared to wt (Fig 4A). Differences were pronounced in the neocortex but also present in subcortical areas. On the contrary, a reduced microglial activity was apparent in the entire cerebrum of $Trem2^{-/-}$ mice, fitting to our recent findings in a TREM2 loss-of-function model (Kleinberger *et al*, 2017; Fig 4A).

Furthermore, we translationally investigated a FTLD patient with a genetically confirmed $GRN$ (NM_002087.2) loss-of-function mutation (c.328C>T, p.(Arg110*)). Haploinsufficiency of PGRN was proven by ELISA-mediated quantitative analysis of the patient's serum (Fig 4B). TSPO-PET imaging revealed elevated cortical microglial activity in comparison with a FTLD patient screened negative for variants in the $GRN$, $MAPT$, $C9orf72$, or the $TREM2$ gene (Fig 4C).

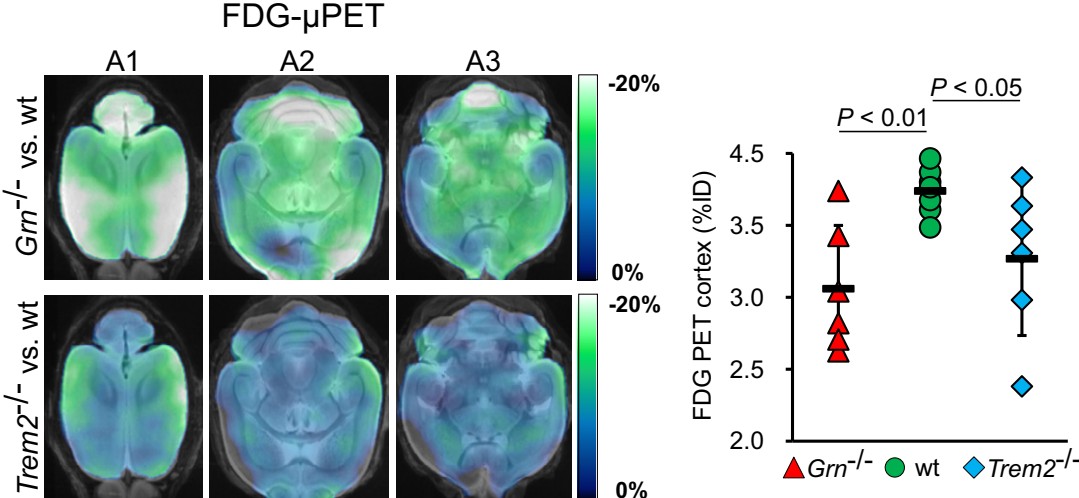

**Figure 5. Cerebral hypometabolism upon loss of GRN and TREM2.**

FDG-µPET in rodents: Axial slices (as indicated in Fig 4A) of averaged %-FDG-µPET differences between $Grn^{-/-}$ or $Trem2^{-/-}$ mice and wt indicate decreased glucose metabolism in the cerebrum of $Grn^{-/-}$ and $Trem2^{-/-}$ (both cold color scales) when compared to wt. Scatter plot depicts single FDG-PET values deriving from a neocortical volume of interest. Data represent the mean +/− SD. µPET results are illustrated upon an MRI template; 6–10 female mice per group ($Grn^{-/-}$ n = 6 at 8 months, $Trem2^{-/-}$ n = 6 at 11 months; wt n = 10 at 8–11 months). Statistics were derived from one-way ANOVA with Tukey post hoc test.

## Cerebral hypometabolism upon loss of PGRN and TREM2

Next, we investigated whether alterations of microglial activity have an impact on synaptic brain function as assessed by *in vivo* FDG-µPET. We observed a cerebral hypometabolism not only in $Trem2^{-/-}$ mice, which had been anticipated from our previous study (Kleinberger *et al*, 2017), but even more severe in $Grn^{-/-}$ mice (Fig 5). As activated microglia itself could lead to increased energy consumption, the results in $Grn^{-/-}$ mice appear even more striking. Thus, a failure of brain function was confirmed as a consequence of both, arresting microglia in a homeostatic and a disease-associated stage.

## Discussion

Taken together, our findings demonstrate that microglia lacking functional TREM2 or PGRN exhibit opposite mRNA profiles. This is consistent with the differential TSPO-PET signal observed in both animal models. While $Grn^{-/-}$ mice exhibit a hyperactivated pheno-type, which is consistent with the observation that $Grn^{-/-}$ mice exhibit an exaggerated inflammation (Yin *et al*, 2010; Martens *et al*, 2012; Tanaka *et al*, 2013a; Lui *et al*, 2016) as well as induced expression of TYROBP network genes including *Trem2* (Takahashi *et al*, 2017), TREM2 loss of function results in reduced microglial activity. As a consequence, several functional assays including phagocytic capacity, migration, and chemotactic clustering around amyloid plaques revealed opposite phenotypes. Thus, loss of PGRN or TREM2 arrests microglia at the two extreme ends of a potential gradient of dynamic microglial populations (Fig 6; upper panel). Nevertheless, in both animal models we observed a dramatic reduction of glucose metabolism throughout the brain and loss of either protein causes severe neurodegeneration in humans. Reduced energy metabolism in $Trem2^{-/-}$ mice is in line with previous

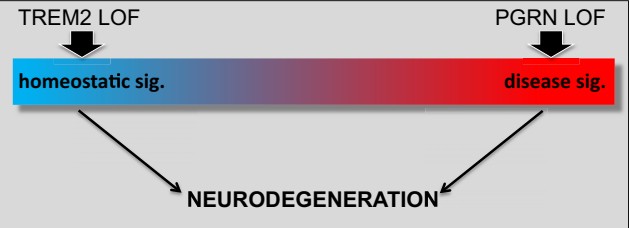

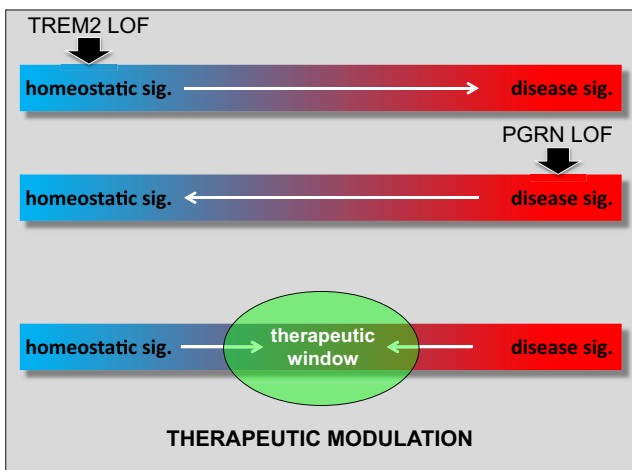

**Figure 6. Modulation of microglia activity states within a narrow therapeutic window.**

Schematic presentation of the disease-associated/homeostatic phenotypes observed in $Grn^{-/-}$ mice versus $Trem2^{-/-}$ and the potential consequences for a therapeutic window of microglial modulation.

observations (Kleinberger *et al*, 2017; Ulland *et al*, 2017). Thus, it is tempting to speculate that microglia on the opposite ends of a large spectrum of functionally divergent microglial populations cause

neurological dysfunctions that ultimately may result in neurodegeneration. In fact, disease-associated variants of TREM2 are associated with a loss of a protective function of TREM2 (Kleinberger *et al*, 2017; Song *et al*, 2018) and some mutations appear to lock microglia in a homeostatic stage (Krasemann *et al*, 2017; Mazaheri *et al*, 2017). On the other hand, during aging wild-type microglia in an AD mouse model lose their motility and phagocytic activity (Krabbe *et al*, 2013) although they express a neurodegenerative disease-associated mRNA profile (Krasemann *et al*, 2017). This may indicate that "over-activation" for a prolonged period of time may be detrimental for microglia and drive them into a loss-of-function phenotype. Our findings also indicate that arresting microglia in either a homeostatic state or a disease-associated state would be equally detrimental. This has important consequences for therapeutic strategies aiming to modulate microglial activity. Clearly, locking microglia in any of the two extreme states is detrimental (Fig 6, upper panel). Rather, microglia locked in a homeostatic state should be activated whereas microglia arrested in a disease-associated state may be silenced (Fig 6, lower panel). Nevertheless, during long-term treatment "over-activation" and "over-silencing" of microglia may both be equally detrimental, since that may drive microglia to the opposite ends of the gradient. As a consequence, the therapeutic window for microglial modulation may be rather limited (Fig 6, lower panel) and care must be taken to balance microglial activity.

# Materials and Methods

### Animal experiments

All animal experiments were performed in accordance with local animal-handling laws. Housing conditions included standard pellet food and water provided *ad libitum*, 12-h light–dark cycle at temperature of 22°C with maximal 5 mice per cage and cage replacement once per week, and regular health monitoring. Mice were sacrificed by $CO_2$ inhalation. Brain tissue, microglia, and serum were obtained from the $Grn^{-/-}$ mouse strain (Kayasuga *et al*, 2007), brain tissue and microglia were obtained from the $Trem2^{-/-}$ mouse strain (Turnbull *et al*, 2006), and brain tissue was obtained from a mouse model for amyloid pathology (APPPS1; Radde *et al*, 2006) crossed with the $Grn^{-/-}$ mouse strain (APPPS1/$Grn^{-/-}$). Experiments were performed with mice of both genders at 2–18 months of age. Organotypic brain slices were taken from postnatal days 6–7 of $Grn^{-/-}$ and wt mice.

### Gene expression profiling

FCRLS (Butovsky *et al*, 2014)- and CD11b (BD biosciences; Clone M1/70)-positive primary microglia were isolated from mouse brains of 5.5 months of $Grn^{-/-}$ and wt mice ($n = 5$ per genotype, male) as described (Mazaheri *et al*, 2017). Total RNA was isolated using mirVanaTM miRNA Isolation Kit (Ambion) according to the manufacturer's protocol. 100 ng RNA per sample was used for gene expressing profiling using nCounter Analysis, NanoString technology as described (Butovsky *et al*, 2014, 2015). In brief, the MG534 NanoString chip was designed using the quantitative NanoString nCounter platform. Selection of genes is based on analyses that identified genes and proteins, which are specifically or highly expressed in adult mouse microglia (Butovsky *et al*, 2014) and encompasses 400 homeostatic unique and enriched microglial genes (Butovsky *et al*, 2014), 129 disease-associated genes, and 5 housekeeping genes (Krasemann *et al*, 2017).

### Data normalization and analysis

NanoString data were normalized and analyzed using nSolver™ software. RNA ncounts were normalized using the geometric mean of five housekeeping genes including *Cltc*, *Gapdh*, *Gusb*, *Hprt1*, and *Tubb5* using nSolver™ Analysis Software, version 3.0 (NanoString Technologies, Inc.). A cutoff was introduced at the value two-fold of the highest negative control present on the chip. Fold changes were calculated using the average of each group. For each experiment, the fold changes were calculated comparing the experimental group to their appropriate controls. To compare the normalized gene expression levels in $Grn^{-/-}$ and wt mice, unpaired two-tailed Student's *t*-test was performed ($*P < 0.05$, $**P < 0.01$, $***P < 0.001$). The volcano blot was generated employing GraphPad Prism 7 software. The heatmap was generated employing Multi Experiment Viewer v 4.9. The expression value for each gene was normalized to the mean value of wt mice followed by a log2 transformation.

### sTREM2 ELISA analysis of mouse plasma and brain samples

Blood collected from $Grn^{-/-}$ and wt mice (3, 6, 13, 16–18 months of age, female and male mice as indicated in the source data file to Fig 2) was left at room temperature (RT) to allow the blood to clot. After centrifugation (15 min, 2,000 × *g*, RT), the supernatant (serum) was collected, snap-frozen in liquid nitrogen, and stored at −80°C.

Snap-frozen mouse brain hemispheres were homogenized by crunching the brain in liquid nitrogen to fine powder. Approximately 10–20 mg of powdered mouse brain from $Grn^{-/-}$ and wt mice (3–4, 6, 13, 16–18 months of age, female and male mice as indicated in the source data file to fig 2) was homogenized in Tris-buffered saline (TBS) supplemented with protease inhibitor cocktail (Sigma-Aldrich) and centrifuged for 30 min, 15,000 × *g*, 4°C. Protein concentrations of supernatants were determined using the BCA protein assay (Pierce, Thermo Scientific), and equal amount of protein was analyzed for TREM2. sTREM2 levels in mouse serum or mouse brain extracts were quantified using the Mesoscale platform essentially as described previously (Kleinberger *et al*, 2017) using 0.25 μg/ml biotinylated polyclonal goat anti-mouse TREM2 (R&D Systems; BAF1729) as capture antibody and 1 μg/ml rat monoclonal anti-TREM2 antibody (R&D Systems, MAB17291) as detection antibody. Calculation of the concentration of sTREM2 was performed with the MSD Discovery Workbench v4 software (MSD).

### Cell culture and CRISPR/Cas9 genome editing in BV2 cells

The murine microglial cell line BV2 (Bocchini *et al*, 1992) was maintained in Dulbecco's modified Eagle's medium (DMEM) + GlutaMAX™- I (Thermo Fisher Scientific, 61965-026) supplemented with 10% (v/v) FBS (Sigma-Aldrich, F7524), 100 U/ml penicillin, and 100 μg/ml streptomycin. For genome editing, BV2 cells were transfected with pSpCas9(BB)-2A-GFP (PX458; gift from Feng Zhang (Ran *et al*, 2013), targeting exons 3 and 4 with the gRNA1 ATAAC GAGCCATCATCTAGA and gRNA2 GGCTTCCACTGTAGTGCAGA;

Addgene plasmid #48138) by electroporation with the Cell Line Nucleofector® Kit T (Lonza VACA-1002) following the manufacturer's instructions. GFP-positive cells were isolated by FACS 24 h after transfection. Single-cell clones from the GFP-positive fraction were obtained by serial dilution and screened for genetic modifications in *Grn* by PCR amplification of exons 3 and 4. Media and lysates of edited clones were analyzed by Western blot for PGRN expression.

## Quantitative real-time PCR (qRT–PCR)

Total RNA was prepared from BV2 cells using the QIAshredder and RNeasy Mini Kit (Qiagen) according to manufacturer's instructions. 2 μg of RNA was reverse transcribed into cDNA using M-MLV reverse transcriptase (Promega) and oligo(dT) primers (Life Technologies). The following primer sets from Integrated DNA Technologies were used: mouse *Trem2* Mm.PT.58.46092560 (Exon boundary 2 to 4b), mouse *Apoe* Mm.PT.58.33516165 (Exon boundary 1 to 3), *Clec7a* Mm.PT.58.42049707 (Exon boundary 1 to 2), and *P2ry12* Mm.PT.58.43542033 (Exon boundary 3 to 4). cDNA levels were measured in triplicates using TaqMan assays on a 7500 Fast Real-Time-PCR System (Applied Biosystems), normalized to *Gapdh* cDNA expression, and relative transcription levels of the respective sequences were analyzed using the comparative delta Ct method (7500 Software V2.0.5, Applied Biosystems, Life Technologies).

## Isolation of adult primary microglia for immunoblotting

Primary microglia were isolated from adult mouse brain (9 months of age, $n = 3$ per genotype, female) using MACS Technology (Miltenyi Biotec) according to manufacturer's instructions as described previously (Gotzl *et al*, 2018).

## Cell lysis and immunoblotting

Snap-frozen cell pellets were lysed in RIPA lysis buffer followed by centrifugation at $17,000 \times g$, 4°C for 20 min [RIPA lysis buffer (150 mM NaCl, 20 mM Tris–HCl pH 7.4, 1% NP-40, 0.05% Triton X-100, 0.5% sodium deoxycholate, 2.5 mM ETDA)]. Protein concentrations were determined using the BCA protein assay (Pierce, Thermo Fisher Scientific), and equal protein amounts were analyzed in SDS–PAGE. For the detection of PGRN, TREM2, CLEC7A, P2RY12, calnexin, CD68, IBA1, and APP gels were transferred on polyvinylidene difluoride membranes (Amersham Hybond P 0.45 PVDF, GE Healthcare Life Science). For the detection of ApoE, nitrocellulose membranes were used (GE Healthcare Life Science). Membranes were blocked for 1 h in I-Block™ (Thermo Fisher Scientific) and exposed to the following antibodies. In-house antibodies were as follows: PGRN (Rat 8H10 1/50; Gotzl *et al*, 2014) and TREM2 (Rat 5F4 1/50; (Xiang *et al*, 2016)). Commercial antibodies were as follows: CLEC7A (R&D Systems, AF1756, 1/2,000), P2RY12 (Abcam, [EPR18611] (ab184411), 1/1,000) and ApoE (Merck, AB947, 1/2,000), and APPs (Merck, MAB348 clone 22C11, 1/5,000). For immunoblotting in primary microglia: ApoE (HJ6.3, 1/700), kindly provided by David E. Holtzman (Kim *et al*, 2012), CD68 (Abcam, ab125212, 1/1,000), and IBA1 (GeneTex, GTX100042, 1/1,000) were used. HRP-conjugated secondary antibodies and ECL

Plus (Pierce™, Thermo Fisher Scientific) were used for detection. For the quantitatively analysis, images were taken by a Luminescent Image Analyzer Vilber/Peqlab Fusion SL and evaluated with the Multi Gauge V3.0 software (Fujifilm Life Science, Tokyo, Japan).

## Phagocytosis assay

Assays to determine the phagocytic capacity were carried out as described before (Kleinberger *et al*, 2014, 2017). Briefly, BV2 cells were seeded at a density of 100,000 cells per well of a 12-well dish and 1 day after seeding, incubated with pHrodo green *E. coli* for 30 and 60 min at 37°C after (pHrodo® Green *E. coli* BioParticles® Thermo Fisher). To analyze *E. coli* uptake, cells were resuspended in FACS buffer (PBS, 2 mM EDTA and 1% FBS) and subjected to flow cytometry. Data were acquired using the MACSQuant®. Cytochalasin D was used as an uptake inhibitor at a final concentration of 10 μM. Phagocytosis assays with primary microglia were performed similar to BV2 cells with the following modifications: Cells were isolated from adult mouse brain (2 months of age, $n = 3$ per genotype, male) as described above. Following isolation, primary microglia were cultivated in DMEM/F-12 (Thermo Fisher Scientific, 31330-038) supplemented with 10% (v/v) FBS (Merck, Sigma-Aldrich), 100 U/ml penicillin, 100 μg/ml streptomycin, M-CSF (10 ng/ml), and TGF-β (50 ng/ml). Four days after seeding at a concentration of 100,000 cells per well of a 24-well plate, cells were incubated with pHrodo for 45 min at 37°C. APC-Cy™7 Rat Anti-CD11b (BD Pharmingen™, M1/70, 557657, 1/100) was used as a cell population control.

## *Ex vivo* migration assay

Organotypic brain slices from postnatal days 6–7 of $Grn^{-/-}$ and wt mice were prepared, cultured, and immunostained as described before (Daria *et al*, 2017; Mazaheri *et al*, 2017). Microglial migration was analyzed after 7 days *ex vivo* as described (Daria *et al*, 2017; Mazaheri *et al*, 2017). Quantification was performed in three independent experiments, each including three biological replicates (three independent slice culture dishes per animal and genotype). From each biological replicate, three defined image regions (775 × 775 μm) of the cortical area in the brain slice were acquired using a Leica SP5 confocal microscope with a 20× dry objective. Tile scan images were captured using a 10× dry objective. The number of migrating CD68-positive microglia and the maximum distance migrated were quantified using Fiji. The migration distance measurements (average of three length measurements per acquired image) were performed on an area defined by two manually drawn lines, one limiting the boarder of the tissue and the other surface covered by at least 90% of migrated microglial cells.

## Immunohistochemistry and image analysis

Following deparaffinization and rehydration of paraffin-embedded brain sections, tissue sections mounted on slides were subjected to citric acid antigen retrieval (1 M sodium citrate in PBS, pH 6.0) and boiled in a microwave for 20 min. After cooling, sections were blocked with 5% donkey serum for 1 h at room temperature. Following this, sections were incubated with primary antibodies against IBA1 (Invitrogen, Thermo Fisher Scientific, 1/300) and β-amyloid

(Biolegend, 6E10, 1/300) over two nights at 4°C. Subsequently, sections were washed and incubated with secondary antibodies (donkey anti-rabbit Alexa Fluor 488, 1/700, donkey anti-mouse Alexa Fluor 555, 1/1,000, respectively) overnight at 4°C. For the P2RY12 co-stainings, brains of 9-month-old female mice (except one male for $Trem2^{-/-}$, $n = 3$ per genotype) were immerse-fixed in 4% PFA for 24 h following perfusion. Brains were subsequently cryoprotected in 30% sucrose for 48 h. After freezing, 50-μm-thick microtome sections were sequentially collected in phosphate buffer saline with 15% glycerol and kept at −80°C until further use. Free-floating sections were blocked in 5% goat serum for 1 h at room temperature and then incubated in primary antibodies P2RY12 (Biolegend, 1/100) and IBA1 (Invitrogen, Thermo Fisher Scientific, 1/300) for two nights at 4°C. Sections were washed and incubated in secondary antibodies (goat anti-rat Alexa Fluor 555, 1/500, goat anti-rabbit Alexa Fluor 488, 1/500) for 3 h at room temperature. Lastly, slides were washed and stained with 4′,6-diamidino-2-phenylindole (DAPI, 5 μg/ml) before mounting coverslips with ProlongTM Gold Antifade reagent (Thermo Fisher Scientific). Images were acquired using a LSM 710 confocal microscope (Zeiss) and the ZEN 2011 software package (black edition, Zeiss). Laser and detector settings were maintained constant for the acquisition of each immunostaining. For analyses, three images were taken per slide using 20× objective (Plan-Apochromat 20×/0.8 M27) at 2,048 × 2,048 pixel resolution, with Z-step size of 1 μm at 16 μm thickness. Approximately 150 plaques per genotype were counted using FIJI software (ImageJ) to calculate the number of IBA1-positive microglia around plaques in the cortex. Microglia coverage was quantified by importing acquired images to FIJI, and data channels were separated (Image\Color\Split Channels). Gaussian filtering was used to remove noise, and intensity distribution for each image was equalized using rolling ball algorithm, which is implemented as background subtraction plug-in in FIJI. For the feasibility of the quantification, all layers from a single image stack were projected on a single slice (Stack\Z projection). Next, the microglia were segmented using automatic thresholding methods in Fiji with "Moments" thresholding setting. Subsequently, the diameter (25 or 50 μm) around each plaque was selected to calculate the sum number of positive pixels over total number of pixels within the selected area. Data for 50 μm were subtracted from 25 μm to calculate the microglia coverage in the periphery of the plaque only. Data were normalized to age-matched APPPS1/$Grn^{+/+}$ or $Grn^{+/+}$ controls (4-month-old mice; two males and two females per genotype).

### Rodent μPET

All rodent μPET procedures followed an established standardized protocol for radiochemistry, acquisition, and post-processing (Brendel et al, 2016; Overhoff et al, 2016). In brief, $^{18}$F-GE180 μPET with an emission window of 60–90 min p.i. was used to measure cerebral microglial activity and $^{18}$F-fluordesoxyglucose (FDG) PET with an emission window of 30–60 min p.i. was used for assessment of cerebral glucose metabolism. We studied $Grn^{-/-}$ mice ($n = 6$; female, 8 months), $Trem2^{-/-}$ mice ($n = 6–8$; female, 11 months), and wt mice ($n = 7–10$; female, 8–11 months) by dual PET. Normalization of injected activity was performed by the previously validated myocardium correction method (Deussing et al, 2017) for TSPO-PET and by percentage of the injected dose (%-ID)

for FDG-PET. Known changes in TSPO and FDG-PET during aging (Brendel et al, 2017) were accounted to compensate for natural age differences between individual mice. Groups of $Grn^{-/-}$ and $Trem2^{-/-}$ mice were averaged and compared against wt mice by calculation of %-differences in each cerebral voxel. Finally, TSPO and FDG-PET values deriving from a cortical target VOI (Rominger et al, 2013) were extracted and compared between groups of different genotype by a one-way ANOVA including Tukey post hoc correction.

### Patient identification, genetic studies, PGRN ELISA, and PET

Written informed consent was obtained from all subjects prior to PET examination, and the experiments conformed to the principles set out in the World Medical Association (WMA) Declaration of Helsinki on ethical principles for medical research and the Department of Health and Human Services Belmont Report.

A female patient, 63 years at date of PET, presented with primary progressive aphasia with semantic and non-fluent features. Genetic testing was performed because of a positive family history (triplet brother with early-onset dementia). Exome sequencing from blood DNA and data analysis was performed as described before (Kremer et al, 2017). In brief, exonic regions were enriched using the SureSelect Human All Exon Kit V6 from Agilent followed by sequencing as 100 bp paired-end runs on an Illumina HiSeq4000 (Illumina, San Diego, CA, USA) to an average read depth of 144x. Reads were aligned to the UCSC human reference assembly (hg19) with Burrows–Wheeler algorithm (BWA v.0.5.9). More than 98% of the target sequences were covered at least 20× in all samples. Single nucleotide variants (SNVs) as well as small insertions and deletions were detected with SAM tools v.0.1.19. In-house custom Perl scripts were used for variant annotation. PGRN serum levels measured by ELISA confirmed the GRN mutation carrier. PGRN levels were measured and quantified using the Mesoscale platform essentially as described previously (Capell et al, 2011) using a biotinylated polyclonal goat anti-human PGRN antibody at 0.2 μg/ml (R&D Systems; BAF2420) as capture antibody and a monoclonal mouse anti-human PGRN antibody at 0.25 μg/ml for detection (R&D Systems, MAB2420). Calculation of the concentration of PGRN was performed with the MSD Discovery Workbench v4 software (MSD). Prior to PET in the patient, the affinity of the TSPO receptor for binding of PET radioligands was assessed by determining the genotype of the SNP rs6971 as previously described (Mizrahi et al, 2012; Owen et al, 2012).

The patient was scanned at the Department of Nuclear Medicine, University hospital of Munich, by an established TSPO-PET protocol (Albert et al, 2017). In brief, PET was acquired with a PET/CT scanner (Biograph 64, Siemens) 60–80 min after intravenous injection of 187 MBq of $^{18}$F-GE180. A low-dose CT scan preceded the PET acquisition and served for attenuation correction. An SUVR image was generated by a global mean scaling and compared visually with TSPO-PET data deriving from a 56-year-old female patient with bvFTD, indicating no mutation in GRN and the same binding affinity status. This patient was scanned secondary to informed consent following the same TSPO-PET scanning procedure. In this patient, mutations in GRN, TREM2, and MAPT were excluded by panel sequencing as well as the C9orf72 hexanucleotide expansion by repeat-primed PCR.

## The paper explained

### Problem

A number of variants in genes selectively expressed in microglia within the brain are associated with an increased risk for neurodegenerative diseases such as Alzheimer's disease and frontotemporal lobar degeneration. Among these genes are progranulin (GRN) and the triggering receptor expressed on myeloid cells 2 (TREM2), which both cause neurodegeneration by a loss of function. Since many functionally different microglial population exist within the brain and since a loss of TREM2 locks microglia in a homeostatic state, we now investigated the mRNA signature of microglia derived from $Grn^{-/-}$ mice and compared the key microglial functions of $Grn^{-/-}$ and $Trem2^{-/-}$ mice.

### Results

While $Trem2^{-/-}$ microglia enhance the expression of genes associated with a homeostatic state, microglia derived from $Grn^{-/-}$ mice showed a massive increase in the disease-associated signature. The opposite mRNA expression profile is reflected by completely divergent functional phenotypes. In contrast to $Trem2^{-/-}$ mice, phagocytic capacity, migration, clustering around amyloid plaques, and TSPO activation were all greatly stimulated upon loss of PGRN in mice. TSPO activation is also detected in a $GRN$-associated FTLD patient. Albeit opposite functional phenotypes, loss of PGRN and TREM2 both reduce cerebral energy metabolism.

### Impact

Opposite microglial phenotypes of PGRN and TREM2 loss-of-function result in similar wide spread brain dysfunction. Thus, microglia arrested at either end of a gradient of functionally divergent microglial populations cause severe neurological syndromes. Importantly, this narrows the therapeutic window in attempts aiming to modulate microglial activity.

## Statistical analysis

All data were analyzed using GraphPad Prism 7. Data are presented as mean ± SD or SEM as indicated. Statistical significance was calculated by unpaired two-tailed Student's $t$-test (Figs 1A–D and 2B, C, D, H, I and J, and 3A, C, D and EV1A and B), one-way ANOVA with Tukey's post hoc test (Figs 2F, 4A and 5) for multiple comparison, and the nonparametric two-tailed Mann–Whitney $U$-test (Figs 3F and EV1C). Statistical evaluations were displayed as follows: n.s. not significant for $P > 0.05$; statistical significance was assumed at *$P < 0.05$; **$P < 0.01$; ***$P < 0.001$; and ****$P < 0.0001$.

For the μPET experiments, animal numbers were calculated for a significance of 0.05 and a power of 80%. Animal numbers were not changed during the experiments, and no outliers were excluded from datasets. For all experiments, the genotype was unknown to the investigators.

## Data availability

The datasets produced in this study are available in the following databases:

- Pathogenic variant in GRN: ClinVar SCV000897721.1 (https://www.ncbi.nlm.nih.gov/clinvar/?term=SCV000897721.1).
- NanoString gene sequence, raw, and processed data: Gene Expression Omnibus GSE129709 (https://www.ncbi.nlm.nih.gov/geo/query/acc.cgi?acc=GSE129709).

Expanded View for this article is available online.

## Acknowledgements

This work was supported by the Deutsche Forschungsgemeinschaft (DFG) within the framework of the Munich Cluster for Systems Neurology (EXC 1010 SyNergy), the Koselleck Project HA1737/16-1, the Cure Alzheimer's fund and the NOMIS Foundation (to C.H.), by a dedicated PET imaging grant to M.B. & A. Ro. (BR4580/1-1 & RO5194/1-1) and the German Federal Ministry of Education and Research (project: FTLDc 01GI1007A; to J.D.-S.), and a PhD stipend of the Hans and Ilse Breuer Foundation to A. Re. We would like to thank Dr. M. Nishihara (Department of Veterinary Physiology, The University of Tokyo) for providing the founder of the $Grn^{-/-}$ mouse strain, Dr. David M. Holtzman (Department of Neurology, Washington University School of Medicine, St. Louis, MO, USA) for providing the ApoE antibody, Dr. Feng Zhang (Broad Institute) for providing PX458 and acknowledge the Core Facility Flow Cytometry at the Biomedical Center, Ludwig-Maximilians-Universität, München, for providing equipment and services and Christian Behrends (Ludwig-Maximilians-Universität, München) for providing assistance with flow cytometry analysis of cultivated primary microglia. The $Trem2^{-/-}$ mice were kindly provided by Marco Colonna (Washington University, School of Medicine), and the APPPS1 colony was established from a breeding pair kindly provided by Mathias Jucker (Hertie-Institute for Clinical Brain Research, University of Tübingen and DZNE-Tübingen). GE made GE-180 cassettes available through an early access model. O.B. is supported by the US National Institutes of Health (NIH) NIH-NINDS (R01 NS088137, R21 NS104609, R21 NS101673), NIH-NIA (R01 AG051812, R01 AG054672), NIH-NEY (R01 EY027921), National Multiple Sclerosis Society (5092A1), Nancy Davis Foundation Faculty Award, Cure Alzheimer's Fund Award, and Amyotrophic Lateral Sclerosis Association and Sanofi.

## Author contributions

CH and AC conceived the study and analyzed the results. CH wrote the manuscript with further input from all co-authors. JKG and OB performed RNA analyses, MB. ARo, MD, and PB performed PET analyses, and LSM and ST performed ex vivo migration assays. GW generated and investigated CRISPR/Cas-modified BV2 cells and performed phagocytosis and Western blot assays on isolated primary microglia, N9 and BV2 cells, KF provided technical support on many experiments. SB assisted with cell sorting. A-VC and ARe performed microglial isolation and qPCR assays. JG and A-VC isolated serum from wt/$Grn^{-/-}$ mice, and GK investigated TREM2 expression in serum and brain of $Grn^{-/-}$ mice. SP provided immunohisto-chemical data on APPPS/$Grn^{-/-}$, $Grn^{-/-}$, and $Trem2^{-/-}$ mice; MW, JL, JW, and JD-S identified FTD patients and performed sequencing analyses. OB, STS, and CM performed the NanoString analysis and helped to interpret the data.

## Conflict of interest

C.H. collaborates with DENALI, participated on one advisory board meeting of Biogen, and received a speaker honorarium from Novartis and Roche. A.R. received speaker honoraria from GE. J.L. is an advisor of Aesku, Axon neuroscience, Hexal and Ionis pharmaceuticals and received speaker fees from Bayer Vital and the Willi Gross Foundation as well as travel reimbursements from AbbVie. M.W. received speaking honoraria from Bayer Vital. O.B. collaborates and consults Sanofi, NanoString, and Cell Signaling Technology. O.B. received speaker honoraria from Sanofi and Amgen.

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
