## [Review Process File · EMBO Molecular Medicine]

Opposite microglial activation stages upon loss of PGRN or TREM2 result in reduced cerebral glucose metabolism

Julia K. Götz, Matthias Brendel, Georg Werner, Samira Parhizkar, Laura Sebastian Monasor, Gernot Kleinberger, Alessio-Vittorio Colombo, Maximilian Deussing, Matias Wagner, Juliane Winkelmann, Janine Diehl-Schmid, Johannes Levin, Katrin Fellerer, Anika Reifschneider, Sebastian Bultmann, Peter Bartenstein, Axel Rominger, Sabina Tahirovic, Scott T. Smith, Charlotte Madore, Oleg Butovsky, Anja Capell & Christian Haass

Review timeline:

Submission date:	22 August 2018
Editorial Decision:	2 October 2018
Revision received:	19 March 2019
Editorial Decision:	4 April 2019
Revision received:	18 April 2019
Accepted:	24 April 2019

Editor: Céline Carret

Transaction Report:

1st Editorial Decision

2 October 2018

Thank you for the submission of your manuscript to EMBO Molecular Medicine. We have now heard back from the two referees whom we asked to evaluate your manuscript.

You will see that both referees find the study interesting and referee 2 is supportive of publication. Referee 1 however suggests additional analyses and data to strengthen the findings. We would like to encourage you to address these comments to the best of your availabilities and please discuss your findings in the context of recent published articles, as recommended.

We would welcome the submission of a revised version within three months for further consideration and would like to encourage you to address all the criticisms raised as suggested to improve conclusiveness and clarity. Please note that EMBO Molecular Medicine strongly supports a single round of revision and that, as acceptance or rejection of the manuscript will depend on another round of review, your responses should be as complete as possible.

I look forward to receiving your revised manuscript.

***** Reviewer's comments *****

Referee #1 (Remarks for Author):

In this article Gotzl et al explore the microglial phenotype and gene expression profile of mice knock out for progranulin gene (GRN) versus TREM2 that both develop neurodegeneration. The main finding of this study is that microglia exhibit opposite gene signatures in both models: While Trem2^{-/-} microglia overexpress an homeostatic signature, Grn^{-/-} microglia overexpress genes associated with neurodegenerative phenotype. They further assess metabolic state of Trem2^{-/-} vs Grn^{-/-} microglia using PET imaging and show that both exhibit reduced glucose metabolism. Although interesting, this study remains very descriptive and correlative, lacking a more and precise thorough analysis as well as validation.

In figure 1B, the authors show specific signature of each model but in order to give a better vision to the reader it would be necessary to show the total number of DEGs obtained for each model (Grn vs Trem2) and the number of DEGs overlapping in the two models. The differences observed here could represent only few% of the total DEGs. It is primordial to give a clearer vision to the reader of how much these two models differs since the results expected would have been that they are very close to each other. Moreover the authors could have compared their data to the signature described in Keren-Shaul et al 2017.

In figure 2: panels C and D, the author stated in the text that P2ry12 mRNA level goes down in the Grn mutant, however how do they explain the absence of downregulation of P2RY12 at the protein level. The authors could have performed a P2ry12 staining on Grn^{-/-} mice microglia. What's about TREM2 and CD11c expression described as neurodegenerative associated markers?

Panel 2E and 2F, verify the color legend. It is confusing!

In the Figure 2 the authors use a cell line called BV2 in order to study Grn mutant form. In order to know if this cell line behaves as the microglia it is requested to perform RNA-Seq of BV2 Grn WT and BV2 Grn Mut and show at the RNA level that the signature obtain on microglia is found in BV2 Grn Mut cell lines.

In Figure 3: The authors investigated the phagocytic capacity of PGRN-deficient BV2 cells. They argue that this cell line recapitulate the expression profile changes found in isolated Grn^{-/-} microglia such as enhanced expression of ApoE, CLEC7A and TREM2. Although interesting, the authors should validate such findings in the primary PGRN-deficient microglia.

Panel 3B, 3C, 3D would benefit from analyzing Trem2^{-/-} mice in parallel at the same time.

Panel 3E, the picture cannot be correctly interpreted since the nuclei staining is missing. Difficult to count the number of IBA1⁺ cells.

Figure 4 and 5 are interesting but very descriptive. They don't help to understand the mechanism explaining how opposite gene signatures with Trem2^{-/-} and Grn^{-/-} induce similar outcome.... The authors do not even explore at all the role of TREM2 in Grn^{-/-} microglia (don't even verify it expression).

On the metabolism aspect of TREM2 deficiency, the authors should discuss the the Ulland et al paper (TREM2 Maintains Microglial Metabolic Fitness in Alzheimer's Disease. Ulland TK, Song WM, Huang SC, Ulrich JD, Sergushichev A, Beatty WL, Loboda AA, Zhou Y, Cairns NJ, Kambal A, Loginicheva E, Gilfillan S, Cella M, Virgin HW, Unanue ER, Wang Y, Artyomov MN, Holtzman DM, Colonna M. Cell. 2017 Aug 10;170(4):649-663.e13. doi: 10.1016/j.cell.2017.07.023.).

Referee #2 (Remarks for Author):

The manuscript by Gotzl et al. describes opposing microglial phenotypes in mouse and BV-2 cell models lacking expression of progranulin (PGRN) or Trem2; however, both models had reduced cerebral glucose metabolism suggesting that at both ends of the microglial activation spectrum, the result is brain dysfunction. The paper is extremely well-written and the data are very clear and straightforward. In general, microglia freshly isolated from Grn^{-/-} mouse brain showed increased expression of microglial genes associated with neurodegeneration (MGnD; e.g., Apoe and Clec7a) whereas, Trem2^{-/-} mouse microglia showed reduced MGnD genes and increase homeostatic genes such as P2ry12. Soluble Trem2 levels were higher in Grn^{-/-} mice and increased with age. Functional in vitro and ex vivo studies revealed that PGRN-deficient BV-2 microglia had increased phagocytic and migratory functions compared to wt BV-2 cells. Similarly, APPPS1/Grn^{-/-} mice had more clustering of microglia around plaques, suggesting that PGRN-deficiency results in microglial hyperactivation. Further confirmation was shown by PET imaging of TSPO tracer, 18F GE180, in mice and 2 humans, one of who has a GRN mutation. GE180 uptake was significantly elevated in Grn^{-/-} mice vs. wt mice and in the FTLN patient with the GRN mutation vs a FTLN patient of similar age and gender without a GRN mutation. Together, these results demonstrate that PRGN loss of function leads to microglial activation. This data (for mice) was compared to previously published Trem2 data and new PET imaging data. In all cases, Trem2 loss of function appears to lock microglia in a homeostatic state thereby reducing their ability to react to local changes in the brain. The authors suggest that these two microglia-related genes represent 2 ends of the spectrum, both of which are deleterious as suggested by the hypometabolism observed in both mouse lines. Therefore, the authors suggest that the therapeutic window for targeting microglia may be very narrow and that maintaining a balance between activity and homeostasis is critical.

Comments:

1. Although the authors present microglial gene expression, phagocytosis, migration and clustering around plaques, is there any change in the morphology of microglia in the various mouse models examined? For example, are the microglia processes different (longer and/or more branched) in the APPPS1/Grn^{-/-} mice vs. APPPS1/Grn^{+/+} mice? Is there less (or lower intensity) P2ry12 staining of microglia in the absence of Grn? Were there more Trem2-positive cells surrounding plaques?
2. Was plaque load similar between APPPS1 mice with and without Grn expression? Based on the in vitro studies, one might expect increased phagocytosis of plaques and therefore, fewer plaques in the PGRN-deficient APPPS1 mice.
3. Was the phagocytic function of Trem2^{-/-} BV-2 cells examined? If not, please describe previous data regarding phagocytic function in Trem2^{-/-} mice or cells.
4. Please add labels to distinguish the 2 groups of genes in Figure 1C (homeostatic on left and MGnD on right).
5. Were male or female wt mice (or both) used for μ PET imaging? (It appears that only females were imaged for the other 2 genotypes.)
6. Age- and amyloid-associated increased uptake of 18F GE180 in mice was first published by Liu, B. et al., J Neuroscience 2015; 35(47):15716-15730. Please include this reference.
7. Please explain "powdered mouse brain" (P10, sTREM2 ELISA). Was the serum and brain derived from male or female mice for this assay?
8. Figure 2: were the samples normalized to the loading control for immunoblotting?
9. Ultimately, what was the effect of overactivation of microglia in Grn^{-/-} mice as they age? Did they show signs of neurodegeneration (pathologically or biochemically)? Did they show any obvious behavioral phenotype? (I do not expect the authors to go back and repeat this study but if they have any information regarding the impact of lifelong overactivation of microglia, it would be of great interest.)

Reviewer 1

In figure 1B, the authors show specific signature of each model but in order to give a better vision to the reader it would be necessary to show the total number of DEGs obtained for each model (Grn vs Trem2) and the number of DEGs overlapping in the two models. The differences observed here could represent only few % of the total DEGs. It is primordial to give a clearer vision to the reader of how much these two models differs since the results expected would have been that they are very close to each other.

As requested, we now show in the **new Fig. 1C** that 355 overlapping genes were detected in both NanoString gene expression analyses from which 38 are significantly changed in both genotypes. 84 % of these genes were regulated in an opposite direction.

Moreover, the authors could have compared their data to the signature described in Keren-Shaul et al 2017.

This has now been done in the first paragraph of the Results section.

In figure 2: panels C and D, the author stated in the text that P2ry12 mRNA level goes down in the Grn mutant, however how do they explain the absence of downregulation of P2RY12 at the protein level.

We have now repeated the western blot analysis using a new highly specific antibody with a broader dynamic range. This allowed the detection of a significant reduction of P2RY12 in PGRN-deficient BV2 cells (see **new Fig. 2 G**). Moreover, we confirm reduced expression of P2YR12 *in vivo* in cortical sections of Grn^{-/-} mice. Furthermore, a substantial increase of P2YR12 positive microglia is observed in cortical sections derived from Trem2^{-/-} mice (see **new Fig. 2 E & F**).

The authors could have performed a P2ry12 staining on Grn-/- mice microglia.

This has been done and is shown in the **new Fig. 2E and F**. In line with our mRNA and protein expression data, microglial P2RY12 expression is reduced in Grn^{-/-} mice but strongly enhanced in Trem2^{-/-} mice.

What's about TREM2 and CD11c expression described as neurodegenerative associated markers?

Trem2 and Cd11c (Itgax) mRNA levels were not significantly altered in Grn^{-/-} microglia (see source data Fig. 1A). However, TREM2 protein levels were enhanced in brain, serum and microglial cell lysates. mRNA levels also increased in the mutant BV2 cells (see **new Fig. 2I**).

Panel 2E and 2F, verify the color legend. It is confusing!

We apologize for the confusing legend. We have corrected that accordingly (see **new Fig. 2 C&D**).

In the Figure 2 the authors use a cell line called BV2 in order to study Grn mutant form. In order to know if this cell line behaves as the microglia it is requested to perform RNA-Seq of BV2 Grn WT and BV2 Grn Mut and show at the RNA level that the signature obtain on microglia is found in BV2 Grn Mut cell lines.

We are aware that BV2 cells are not microglia (Butovsky, Nature Neuroscience 2014) and described them accordingly as “microglia-like”. However, important markers of MGnD and HM are indeed correctly expressed in BV2 cells. For the disease associated microglia we show mRNA and protein expression data for ApoE, CLEC7A, TREM2 as well as new data for P2YR12 (see **new Fig. 2G & H**). Moreover, these data are now fully confirmed in primary microglia (see **new Fig. 2A & B**).

In Figure 3: The authors investigated the phagocytic capacity of PGRN-deficient BV2 cells. They argue that this cell line recapitulate the expression profile changes found in isolated Grn-/- microglia such as enhanced expression of ApoE, CLEC7A and TREM2. Although interesting, the authors should validate such findings in the primary PGRN-deficient microglia.

Phagocytosis of *Trem2*^{-/-} microglia is reduced but only slightly enhanced in microglia isolated from *Grn*^{-/-} mice (see **new EV Fig. 1B**). This is likely due to M-CSF and TGF- β treatment and activation of microglia during the culturing conditions (see also Gosselin et al., Science, 2017). Cultured microglia are therefore already activated making it difficult to detect an even further activation in the *Grn*^{-/-} microglia.

Panel 3B, 3C, 3D would benefit from analyzing Trem2^{-/-} mice in parallel at the same time.

We have published migration assays in the ex vivo model for the TREM2 loss-of-function before (Mazaheri et al., EMBO Reports 2017), we therefore refer to these findings.

Panel 3E, the picture cannot be correctly interpreted since the nuclei staining is missing. Difficult to count the number of IBA1+ cells.

We have now repeated the immunohistochemistry and included nuclear staining. This is shown in the **new Fig. 3E and F** with additional quantifications shown in the new **EV Fig. 1C**.

Figure 4 and 5 are interesting but very descriptive. They don't help to understand the mechanism explaining how opposite gene signatures with Trem2^{-/-} and Grn^{-/-} induce similar outcome....

We have intensively discussed how “over activation” similar to “over silencing” may result in detrimental outcomes. The mechanistic explanation is that microglia are locked on either end of the phenotypic spectrum characterized by signifying molecular signatures resulting in opposite functional phenotypes. However, microglia need to flexibly respond to environmental cues, which in both cases is not possible anymore.

The authors do not even explore at all the role of TREM2 in Grn^{-/-} microglia (don't even verify it expression).

Expression of *Trem2* on mRNA level in microglia was analyzed. We now additionally investigated TREM2 expression in acutely isolated microglia from *Grn*^{-/-} mice and demonstrate increased levels of immature and mature TREM2 protein (see **new Fig. 2A & B**). The role of TREM2 in *Grn*^{-/-} mice is now investigated in double knockouts, however, this is outside of the scope of the current manuscript.

On the metabolism aspect of TREM2 deficiency, the authors should discuss the the Ulland et al paper (TREM2 Maintains Microglial Metabolic Fitness in Alzheimer's Disease. Ulland TK, Song WM, Huang SC, Ulrich JD, Sergushichev A, Beatty WL, Loboda AA, Zhou Y, Cairns NJ, Kambal A, Lognischeva E, Gilfillan S, Cella M, Virgin HW, Unanue ER, Wang Y, Artyomov MN, Holtzman DM, Colonna M. Cell. 2017 Aug 10;170(4):649-663.e13. doi: 10.1016/j.cell.2017.07.023.).

This paper has now been discussed as requested.

Reviewer 2

*1. Although the authors present microglial gene expression, phagocytosis, migration and clustering around plaques, is there any change in the morphology of microglia in the various mouse models examined? For example, are the microglia processes different (longer and/or more branched) in the APPPS1/*Grn*^{-/-} mice vs. APPPS1/*Grn*^{+/+} mice? Is there less (or lower intensity) P2ry12 staining of microglia in the absence of *Grn*?*

We have now investigated P2RY12 staining in cortices of wt, *Grn*^{-/-} and *Trem2*^{-/-} mice. Indeed, *Grn*^{-/-} mice showed reduced P2RY12 staining, whereas *Trem2*^{-/-} mice showed a significant increase of P2RY12 (see **new Fig. 2C & D**). However, it was not possible to investigate microglial morphology with the brain sections available.

*2. Was plaque load similar between APPPS1 mice with and without *Grn* expression? Based on the in vitro studies, one might expect increased phagocytosis of plaques and therefore, fewer plaques in the PGRN-deficient APPPS1 mice.*

We did not observe obvious differences in the amyloid plaque load in aged mice.

Of note, previous analyses of amyloid plaque load in PGRN deficient mice revealed discrepant results (Minami et al., Nature Medicine 2014; Pereson et al., J. Pathology, 2009; Van Kampen and Kay, PloS one 2017; Revuelta et al., A. J. of Alzheimer's Disease & other disorders 2008; Hosokawa et al., Exp. Anim. 2015; Takahashi et al., Acta Neuropath. 2017). Based on our recent publication (Parhizkar et al., Nature Neuroscience 2019), investigation of newborn plaques in seeding experiments would be the best way to study effects of microglia on amyloid plaque load as ageing seems to influence TREM2 dependent microglial activity. However, this is outside of the scope of this manuscript.

3. Was the phagocytic function of Trem2^{-/-} BV-2 cells examined? If not, please describe previous data regarding phagocytic function in Trem2^{-/-} mice or cells.

We have previously described the reduced phagocytic function upon TREM2 loss-of-function (Kleinberger et al., Science Translational Medicine 2014). We now added additional data showing the reduced phagocytic activity of primary Trem2^{-/-} microglia and TREM2-deficient N9 cells in the new EV Fig. 1 A & B.

4. Please add labels to distinguish the 2 groups of genes in Figure 1C (homeostatic on left and MGnD on right).

We added the labels as requested.

5. Were male or female wt mice (or both) used for μPET imaging? (It appears that only females were imaged for the other 2 genotypes.)

Only female mice were used for all PET imaging experiments. We included the additional gender information in the method section and in the figure legend of Fig 4 A and Fig 5.

6. Age- and amyloid-associated increased uptake of 18F GE180 in mice was first published by Liu, B. et al., J Neuroscience 2015; 35(47):15716-15730.

We are now referring to this manuscript.

7. Please explain "powdered mouse brain" (P10, sTREM2 ELISA).

Snap frozen mouse brain hemispheres were homogenized by crunching the brain in liquid nitrogen to fine powder. This is now described in more detail in the Materials and Methods section.

Was the serum and brain derived from male or female mice for this assay?

Serum and brain samples for the sTREM2 ELISA were derived from both, female and male mice as indicated in the **new source table** to Fig. 2C & D.

8. *Figure 2: were the samples normalized to the loading control for immunoblotting?*

In all cases protein concentrations were measured with the BCA assay and equal amounts of protein were loaded. The “loading control” served as an additional control to verify equal protein loading.

9. *Ultimately, what was the effect of overactivation of microglia in Grn^{-/-} mice as they age? Did they show signs of neurodegeneration (pathologically or biochemically)?*

There are no obvious signs of neurodegeneration (as in many models for other neurodegenerative diseases such as AD), however, abnormalities in lysosomal function, lipofuscin accumulation, and TDP-43 phosphorylation were described (e.g. Kleinberger et al., J. Neurochem., 2010; Yin, F., et al., J. Exp. Med 2010, Ahmed, Z., et al., Am. J. Pathol. 2010, Götzl et al., Acta Neuroptho. 2014; Götzl et al., Mol. Neurodeg. 2018). In addition, reduced spine density was described (Petkau et al., Neurobiol. of Disease 2012).

Did they show any obvious behavioral phenotype? (I do not expect the authors to go back and repeat this study but if they have any information regarding the impact of lifelong overactivation of microglia, it would be of great interest.)

If the observed behavioral changes such as impaired spatial learning (Yin et al., FASEB J 2010) and abnormalities in aggression, anxiety, sexual and social behavior (Kayasuga et al., Behav. Brain Res. 2007, Filiano et al., J. Neurosci 2013, Ghoshal et al., Neurobiol. Dis. 2012, Arrant et al., Brain 2017) are related to microglial dysfunction remains unclear, although Lui and colleagues link the excessive grooming behaviors in aged Grn knockout mice to enhanced microglial activity associated with increased synaptic pruning (Lui et al., Cell 2016).

2nd Editorial Decision

4 April 2019

Thank you for the submission of your revised manuscript to EMBO Molecular Medicine. We have now received the enclosed report from the referee who was asked to re-assess it. As you will see the reviewer is now supportive and I am pleased to inform you that we will be able to accept your manuscript pending minor editorial amendments.

I look forward to seeing a revised form of your manuscript.

***** Reviewer's comments *****

Referee #1 (Remarks for Author):

The authors revised adequately the manuscript.

2nd Revision - authors' response

18 April 2019

Authors made the requested editorial changes.

Corresponding Author Name: Anja Capell and Christian Haass

Manuscript Number: EMM-2018-09711-V2